# Employment status before and after open heart valve surgery: A cohort study

Britt Borregaard[1,2,3,4]*, Jordi S. Dahl[3,4], Ola Ekholm[5], Emil Fosbøl[6], Lars P. S. Riber[1,3], Kirstine L. Sibilitz[6], Sasja M. Pedersen[7], Thomas P. H. Rothberg[7], Maiken H. Nielsen[3,4], Selina K. Berg[6,8], Jacob E. Møller[3,4,6]

1 Department of Cardiothoracic and Vascular Surgery, Odense University Hospital, Odense, Denmark, 2 OPEN, Odense Patient Data Explorative Network, Odense University Hospital, Odense, Denmark, 3 Faculty of Health Sciences, University of Southern Denmark, Odense, Denmark, 4 Department of Cardiology, Odense University Hospital, Odense, Denmark, 5 National Institute of Public Health, University of Southern Denmark, Copenhagen, Denmark, 6 The Heart Centre, Rigshospitalet, Copenhagen University Hospital, Copenhagen, Denmark, 7 Faculty of Business and Social Sciences, University of Southern Denmark, Odense, Denmark, 8 Department of Clinical Medicine, Faculty of Health and Medical Sciences, University of Copenhagen, Copenhagen, Denmark

* britt.borregaard@rsyd.dk

**Data Availability Statement:** The datasets generated and/or analysed during the current study are not publicly available due to Danish Legislation. The data underlying the results presented in the study are available from corresponding author after

## Abstract

### Objective

Detachment from the workforce following open heart valve surgery is a burden for the patient and society. The objectives were to examine patterns of employment status at different time points and to investigate factors associated with a lower likelihood of returning to the workforce within six months.

### Methods

A cohort study of patients aged 18–63 undergoing valvular surgery at a Danish tertiary centre from 2013–2017. Return to the workforce was defined as being employed, unemployed (still capable of working) or receiving paid leave of absence. The association between demographic-, clinical characteristics (including a surgical risk evaluation, EuroScore), and return to the workforce were investigated with a multivariable logistic regression model.

### Results

In total, 1,395 consecutive patients underwent surgery, 347 were between 18 and 63 years and eligible for inclusion. Of those, 282 were attached to the workforce before surgery and included in the study. At the time of surgery, 79% were on paid sick leave. After six months, 21% of the patients (being part of the workforce before surgery), were still on sick leave. In the regression model, prolonged sick leave prior to surgery (OR 0.43, 95% CI 0.23–0.79) and EuroScore ≥ 2.3 (OR 0.39, 95% CI 0.21–0.74) significantly reduced the likelihood of returning to the workforce.

approval by the Danish Data Protection Agency department that handles data access for Odense University Hospital, which can be contacted at ouh. pd@rsyd.dk. Corresponding author Britt Borregaard can be contacted at britt. borregaard@rsyd.dk.

**Funding:** Helsefonden, The Odense University Hospital PhD foundation, Ove William Buhl Olesen and Edith Buhl Olesen Foundation and Kurt Bønnelycke and Grethe Bønnelycke Foundation funded this research. The funders had no role in either concept, design or interpretation of the study results.

**Competing interests:** The authers have declared that no competing interests exist.

## Conclusion

One-fifth of patients in the working-age were on sick leave six months after surgery. Prolonged sick leave prior to surgery and a EuroScore $\geq$2.3 were associated with a lower likelihood of returning to the workforce.

## Introduction

Among patients capable of working and undergoing cardiac surgery, factors influencing sick leave, how to recover and how to return to work are essential parts of the pathway to recovery after discharge and thus, important outcomes to measure [1]. Prolonged sick leave may have individual patient consequences by impacting the ability to resume daily living in the period after surgery. Furthermore, sick leave and prolonged time to return to the workforce after a treatment is a genuine societal problem with substantial economic consequences for the patient as well as society [2].

Return to the workforce is influenced by a person's health and workability [3], but likewise, other factors might impact the ability to return to the workforce following open heart valve surgery. As the early period after discharge is a seemingly vulnerable period, the patients have an increased risk of experiencing complications and subsequent unplanned readmissions [4–7]. Also, patients experience symptoms of anxiety, depression, reduced quality of life and a changed bodily awareness in the early period after surgery [8–11]. Together, these factors might also influence the return to the workforce after surgery.

Knowledge of specific factors associated with return to the workforce after open heart valve surgery is sparse: Unemployment one year before surgery has been associated with reduced likelihood of returning to the workforce [1], whereas participating in cardiac rehabilitation is not associated with returning to the workforce [12]. Some studies have investigated factors associated with the likelihood of returning to the workforce following coronary artery bypass grafting (CABG) and demonstrated how younger age, male sex, higher educational level, work status before surgery and higher income were associated with returning to the workforce [1, 13, 14]. Also, sick leave status before treatment and job type/strain are known to impact employment status following a heart disease [15]. Whether similar characteristics, including differences in age groups, are associated with returning to the workforce among patients undergoing open heart valve surgery, are currently unknown. This knowledge is vital to identify patients at higher risk and thereby to prevent detachment from the workforce to the extent possible.

Thus, in a population of patients undergoing open heart valve surgery, the objectives of the study were to i) examine patterns of employment status at different time points, including differences among age groups and to ii) investigate demographic and clinical factors associated with a lower likelihood of returning to the workforce within six months after surgery.

## Materials and methods

The current study is an exploratory cohort study investigating employment status six months before and after open heart valve surgery.

### Participants and setting and recruitment

Patients undergoing open heart valve surgery at a high volume tertiary centre, Odense University Hospital, Denmark, from August 2013 to November 2017 were consecutively included in

the cohort study [7]. Open heart valve surgery was defined as one of the following surgical procedures (Nordic/NOMESCO Classification of Surgical Procedures [16]): Aortic (KFCA, KFMA, KFMC, KFMD), Mitral (KFKB, KFKC, KFKD, KDKW) and Tricuspid (KFGC, KFGE).

Excluded were: patients $\geq$ 64 years (n = 1,027), patients who died during index admission *or* within 180-days after discharge (n = 12), patients who required endocarditis treatment during the index admission due to prolonged hospitalisation (n = 8) and patients who developed perioperative stroke and received neurological rehabilitation (n = 1) and patients who did not have a Danish civil registration number due to lack of data on work status (n = 2). Thus, the current study was restricted to patients at the working age (between 18 and 63 years) at inclusion time to ensure full follow-up before the patients were offered state pension (at the age of 65 years during the study period).

## Data collection

**Demographic and clinical data.** Demographic and clinical data were obtained from electronic medical records and the Western Denmark Heart Registry (WDHR) [17]. Living status, smoking status, alcohol consumption, body mass index (BMI) and length of stay were obtained from the electronic medical records, whereas the type of surgery, comorbidity and EuroScore II (surgical risk evaluation) were obtained from WDHR. The EuroScore II is a logistic surgical risk evaluation calculated before surgery including age, sex, renal impairment, extracardiac arteriopathy, poor mobility, chronic lung disease, active endocarditis, critical preoperative state, angina status, recent myocardial infarction, pulmonary hypertension, urgency and weight of the procedure [18].

**Sick leave and workforce attachment.** In Denmark, all citizens are entitled to social security benefits during sick leave. Data on sick leave and labour marked absence were acquired from the Danish DREAM registry, administered by the Danish Ministry of Employment. The DREAM registry contains information on social security benefits of all individuals who have received social transfer payment and consists of more than 100 codes on benefits reported with weekly status [19, 20]. Only sick leave lasting for more than two weeks is included in the registry, meaning that the status of working citizens based on short-term sick leave is not registered [19, 20].

As all citizens in Denmark have a unique, national civil registration number, the population of the study were matched to the DREAM registry based on the civil registration number ensuring that data on each patient was matched to specific sick leave benefits. Variables in the DREAM registry were grouped based on recommendations and similar study designs [13, 21, 22] related to the workforce attachment into:

1. On sick leave

2. Working/part of the workforce (employed, unemployed (but still capable of working based on the coding) or received paid leave of absence and educational grants)

3. Out of the workforce (patients were considered to be out of the workforce if they were on early retirement of any kind, n = 65)

Employment status six months prior to admission for the open heart valve surgery was determined based on being capable of working during the period. To reduce misclassification, work status was evaluated based on the status of each week before surgery by ensuring that patients who had a short term sick leave period were not grouped as being detached from the workforce [13]. Patients who were part of the workforce for at least two weeks (consecutive) in

the period before surgery were included in the study. Also, patients on prolonged sick leave prior to surgery were defined as patients $\geq$ 2 weeks of sick leave.

## Outcomes

Study outcomes included the return to the workforce six months after surgery and employment status (sick leave, part of the workforce or out of the workforce) before and up to six months after open heart valve surgery.

## Statistics

Baseline characteristics were presented as mean and standard deviation (SD) or as median and 25th to 75th percentiles (IQR) for continuous data and number and percentage for categorical data. Normality was tested with the Shapiro-Wilks test.

Distribution of sick leave six months before and after surgery were visualised with a probability distribution plot (proportion of the population) for the overall population and divided into age groups. Differences in the proportion of patients on sick leave, part of the workforce and out of the workforce at different timepoints were described.

To investigate factors associated with returning to the workforce within six months after surgery, logistic regression analyses were performed. A multiple logistic regression model was chosen, as only two patients died during the 180-days follow-up. The following variables were included in the model; sex, age groups, surgical procedure, prolonged sick leave before surgery (defined as sick leave $\geq$ 2 weeks before surgery), EuroScore log II, postoperative atrial fibrillation and length of stay. The included variables were assumed to be associated with the outcome, but with restricted numbers to avoid overfitting the model. Model fit was tested with the Likelihood Ratio.

As a sensitivity analysis, we removed patients receiving educational grants from the overall regression analysis.

A P-value of <0.05 was considered to be statistically significant. SPSS 24 (IBM Corp, Armonk, NY) and R 3.2.2 (R Foundation for Statistical Computing) were used for the analyses.

## Ethics approval

The study was approved by the Danish Data Protection Agency (18/19152), the Danish Patient Safety Authority and conformed with the principles outlined in the Declaration of Helsinki [23]. Due to Danish Legislation, signed consent for register-based studies is not needed.

## Results

In total, 1,395 patients underwent open heart valve surgery during the period of which 368 patients were between 18 and 63 years, and thus in a working age. Of those, 347 were eligible after the exclusion criteria and withdrawal of patients on early retirement, n = 282 were included in the study (Fig 1). Clinical and demographic characteristics among patients being part of the workforce are summarised in Table 1 and among the total population in S1 Table. The majority of the patients being part of the workforce were men (81%), the median age was 55 years (IQR 49–60), and 26% were living alone at the time of surgery. A total of 124 patients were diagnosed with aortic valve stenosis (44%), and concomitant coronary artery bypass grafting (CABG) was performed in 9%. In addition, prior to the surgery, 42% had a prolonged sick leave period (sick leave $\geq$ 2 weeks), Table 1.

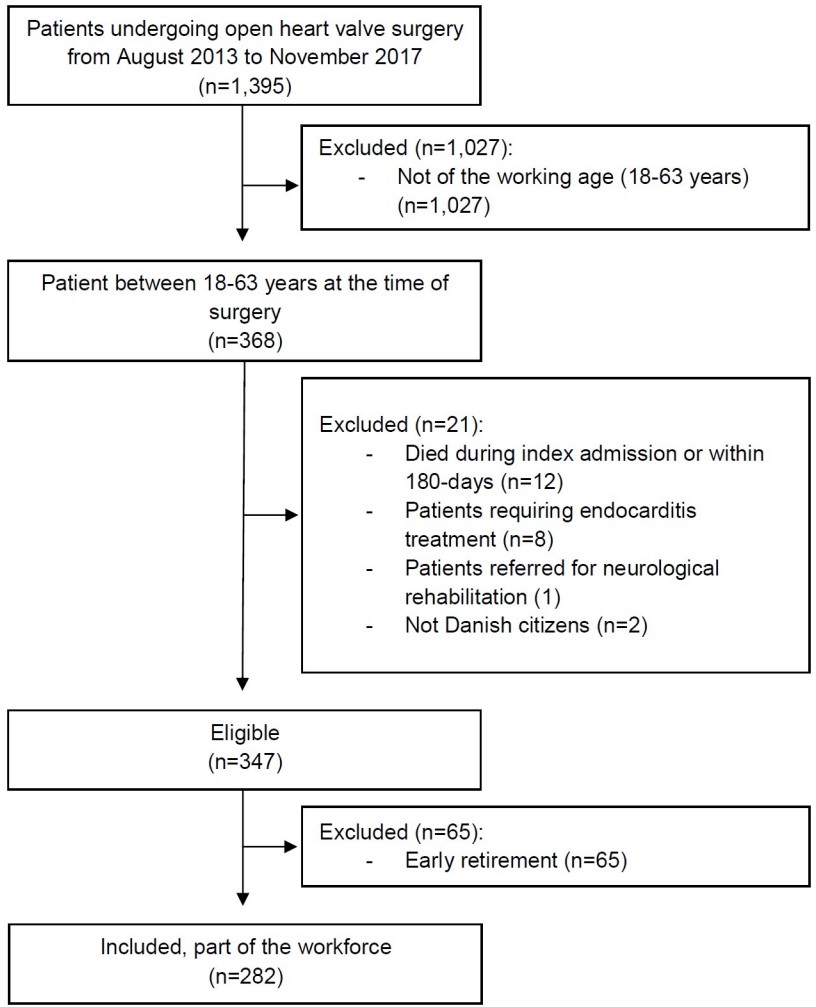

**Fig 1. Patient flowchart.** Flowchart of the patient population.

## The pattern of employment status and return to the workforce after open heart valve surgery

The pattern of the employment status (being on sick leave or working) before and after surgery among the total population of patients ≤ 63 years are showed in Fig 2.

Patterns of sick leave among the patients being part of the workforce (n = 282) varied at different time points, with n = 7 (2%) being on paid sick leave six months before surgery and n = 223 (79%) receiving sick leave benefits at the time of surgery. The proportion of patients being on paid sick leave at the time of surgery varied across the different age groups, with 67% of the patients in the age group from 18–45 years being on sick leave versus 84% of the patients in the age group from 56–63 years (S2 Table).

At six months after surgery, most patients being part of the workforce before surgery had returned to the workforce, although n = 59 (21%) were still on sick leave (S2 Table). The highest proportion of patients who did not return to the workforce were found in the youngest age group (age 18–45) with 29% still being on sick leave six months after surgery (Fig 3 and S2 Table). Also, n = 6 (2%) had left the workforce after six months. Median time on sick leave during the total period was 15 weeks (IQR 8;24).

**Table 1. Baseline characteristics of the patients being part of the workforce.**

|  | Part of the workforce (n = 282) |
|---|---|
| **Characteristics** |  |
| Sex, male, n (%) | 229 (81) |
| *Age-groups, n (%)* |  |
| 18–45 years | 42 (15) |
| 46–50 years | 46 (16) |
| 51–55 years | 55 (20) |
| 56–63 years | 139 (49) |
| Living alone, n (%) | 72 (26) |
| Sick leave ≥ 2 weeks before surgery | 120 (42) |
| **Pre-operative information** |  |
| Reduced pulmonary function[a], n (%) | 93 (33) |
| EuroScore II (logistic), median (IQR) | 1.15 (0.73–2.44) |
| EuroScoreII ≥2.3, n (%) | 73 (26) |
| Hypertension (medical treatment) | 192 (68) |
| Estimated glomerular filtration rate ml/min.[b], median (IQR) | 101 (83–123) |
| Atrial fibrillation, n (%) | 46 (16) |
| Diabetes[c], n (%) | 24 (9) |
| Ejection fraction ≤50%, n (%) | 78 (28) |
| NYHA class ≥3, n (%) | 86 (31) |
| Body Mass Index, median (IQR) | 26 (24–30) |
| Current or former smoker, n (%) | 140 (50) |
| Alcohol intake above national recommendations, n (%) | 30 (11) |
| *Primary diagnosis, n (%)* |  |
| Aortic valve stenosis | 124 (44) |
| Aortic valve regurgitation | 72 (26) |
| Mitral valve stenosis | <5 (1) |
| Mitral valve regurgitation | 83 (29) |
| **Surgical information, n (%)** |  |
| *Type of valve procedure*[d] |  |
| Aortic valve, biological | 37 (13) |
| Aortic valve, mechanical | 152 (54) |
| Aortic valve, repair | 6 (2) |
| Mitral valve, replacement[e] | 21 (7) |
| Mitral valve, repair | 65 (23) |
| Concomitant CABG | 24 (9) |
| **Post-procedure related, n (%)** |  |
| Re-operation | 15 (5) |
| Prolonged length of stay[f], intensive care unit | 28 (10) |
| Postoperative atrial fibrillation | 132 (47) |
| New-onset postoperative atrial fibrillation | 99 (35) |
| *Length of stay* |  |
| 4–7 days | 103 (37) |
| 8–12 days | 106 (38) |

(*Continued*)

**Table 1.** (Continued)

|  | Part of the workforce (n = 282) |
| --- | --- |
| ≥13 days | 73 (26) |

IQR, interquartile range, 25[th] to 75[th] quartile. NYHA, New York Heart Association Class.

[*] The total population of patients between 18–63 years.

[a] Patients with forced expiratory volume, % ≤80% of predicted value and / or a history of chronic obstructive pulmonary disease.

[b] Estimated glomerular filtration rate estimated by the Cockcroft-Gault Equation.

[c] Patients with diabetes; insulin, peroral and non-pharmacological treatment.

[d] One patient had surgery on the tricuspid valve and are not shown in the table, but included in the analyses.

[e] Both biological and mechanical mitral valve replacement.

[f] Admission at the intensive care unit for more than one day.

The pattern of the overall employment status (being on sick leave, working or being out of the workforce) before and after surgery among the total population of patients ≤ 63 years are shown in S1 Fig and S3 Table. At the time of surgery, 19% of the total population were out of the workforce (S2 Table).

## Factors associated with return to the workforce within six months after discharge

In the multiple logistic regression model, prolonged sick leave prior to surgery (OR 0.43, 95% CI 0.23–0.79) and EuroScore ≥ 2.3 (OR 0.39, 95% CI 0.21–0.74) significantly reduced the likelihood of returning to the workforce (Fig 4).

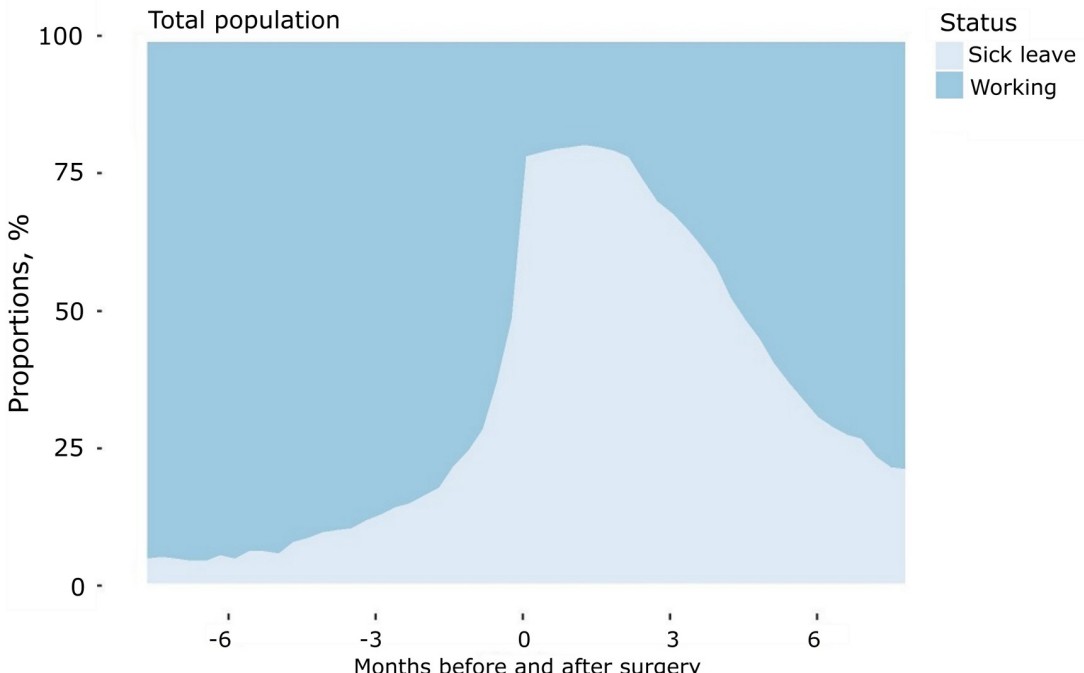

**Fig 2. The proportional distribution of employment status six months before and after surgery among patients between 18–63 years.** The figure illustrates the status of the patients at different time points before and after surgery. Status of patients who are part of the workforce and patients on sick leave are visualised.

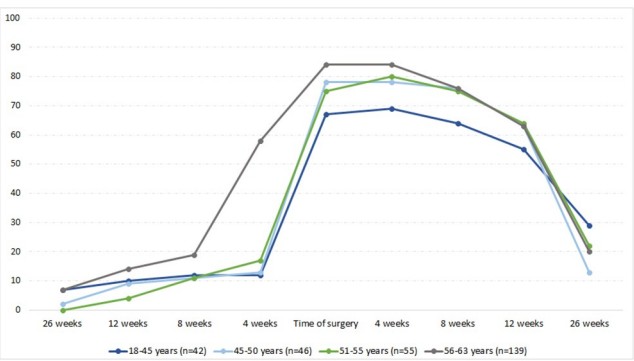

**Fig 3. The proportion of patients being on sick leave at different time points, divided into age groups.** The figure illustrates the status of the patients (on sick leave) at different time points before and after surgery, divided into age groups.

The sensitivity analysis where patients receiving educational grants were removed did not change the overall results (S2 Fig).

## Discussion

In this cohort study, we investigated return to the workforce and factors associated with returning to the workforce within six months after open heart valve surgery. At the time of surgery, a lower proportion of patients in the youngest age group were on paid sick leave compared to the other age groups. Also, one-fifth of the patients were still on paid sick leave six months after surgery. Prolonged sick leave prior to surgery and a EuroScore $\geq$ 2.3 significantly reduced the likelihood of returning to the workforce.

Earlier studies have focused on different aspects of return to the workforce following open heart valve surgery: work status before surgery and participation in cardiac rehabilitation and the likelihood of returning to the workforce [1, 12]. Thus, to our knowledge, this is the first study to investigate clinical characteristics and the association with return to the workforce. As expected, the proportion of patients being on paid sick leave reached its maximum level at the time of surgery, but unexpected, fever patients in the younger age group (from 18–45 years) were on paid sick leave at that time point. One possible explanation for this might be due to more patients in the young group being studying/receiving educational grants, as previously demonstrated in a young population of patients with endocarditis [22]. Students in Denmark receive a paid monthly fee (educational grants) by the government, but do not receive paid sick leave. Similarly, more patients in the young group received immigration fees which also meant, that they did not receive paid sick leave. As the total number of patients in the young group is low, the few patients receiving educational grants or immigration fee (14% of the patients in the young group) might have an impact on the number on paid sick leave. Removing these patients did not change the results of the regression analysis.

Interestingly, 17% of the total population did not receive any paid sick leave at the time of the surgery. This does not necessarily imply that the patients had a working status, but only, that they did not receive paid sick leave as part of the national social security system. The results comply with an earlier Danish register-based study by Fonager et al., where 16% of a similar study population did not receive paid sick leave at the time of surgery [1]. Fonager et al. speculate how some patients might be economically funded by their spouses (although this might only be a seemingly low proportion), but also how the registration in the Danish

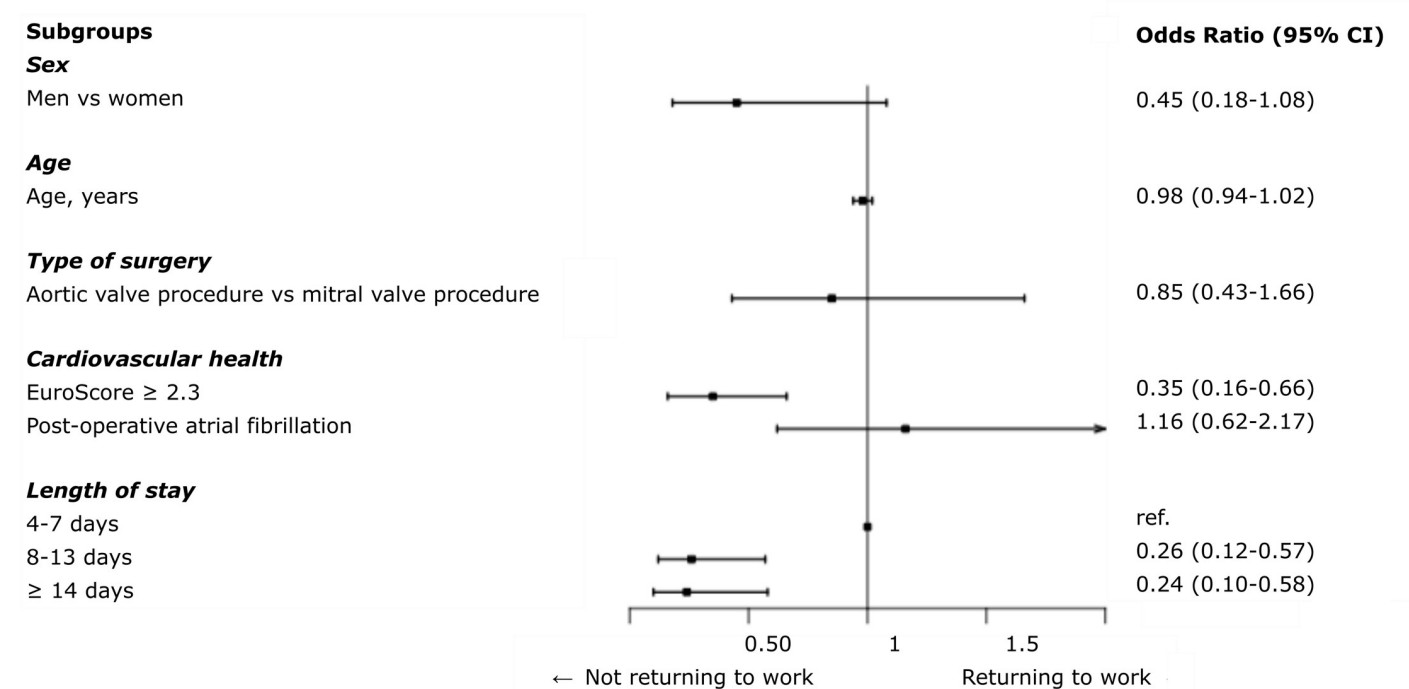

**Fig 4. Factors associated with returning to the workforce.** The figure illustrates the logistic regression model of factors associated with returning to the workforce, with time to return to work (weeks) as the underlying time scale.

DREAM database with requirements of two weeks sick leave might impact their results [1]. In addition, patients who are self-employed are not necessarily being on sick leave, depending on the status of their business.

At the six months follow-up, we demonstrated that one-fifth of the patients were still on sick leave which is similar to earlier studies among patients undergoing open heart valve surgery or patients undergoing CABG [1, 13].

In the regression analyses, prolonged sick leave prior to surgery and EuroScore ≥2.3 were associated with a lower likelihood of returning to the workforce. Thus, as expected, returning to the workforce was dependent on pre-operative morbidity (seen as prolonged sick leave) and surgical risk. Although tested in a small population in the current study, the results comply with a study among patients undergoing CABG; a high comorbidity burden is associated with a lower likelihood of returning to the workforce [13].

In general, research investigating the pattern of sick leave and factors of return to the workforce after both open heart valve surgery and comparable populations are sparse and needs to be investigated more thoroughly as there are several personal- and economic advantages for both patients and society in returning to the workforce. Similarly, to prevent detachment from the workforce to the extent possible, a better understanding of the associated factors is warranted. Currently, research investigating interventions aiming at increasing the likelihood of returning to the workforce is sparse with current studies lacking statistical power or reporting conflicting results [12, 24].

## Strengths and limitations

The use of the DREAM register is both the main strength and a limitation of the study; A strength as it covers all entries on sick leave compensations. As the entries are dependent on

employers claiming compensation refund, the incentive for registration is high–ensuring high accuracy of data. The use of the register though, also includes limitations as only sick leave spells of >2 weeks are included. Thus, the register does not include short-term absence, and similarly, the register does not distinguish between full- and part-time sick leave [19, 20]. Similarly, as the register only include persons who receive a paid benefit, it can be difficult to conclude on the status of persons who are not registered. Type of job, job strain (e.g. physical strenuous fields), self-employment and similar knowledge would potentially also be relevant data of the population, but not available through the DREAM registry. This is also a limitation by the use of the DREAM register. The number of available jobs during the study period could potentially have had an impact on returning to work, why the main focus of the study was returning to the workforce. Classification of the groups; sick leave, workforce and out of the workforce could have caused a risk of classification bias but were performed as a method to classify DREAM data as previously described [13, 21, 22]. In addition, the analyses were performed based on a small population which might cause a problem in the regression analyses due to possible lack of statistical power.

Exclusion of patients with endocarditis and patients requiring neurological rehabilitation cause another limitation of the study. Patients with infective endocarditis have a hospital course very different from elective heart valve surgery; a common need for prolonged antibiotic treatment and often complications related to the systemic infection contrary to the surgery. Thus, these patients were excluded. In addition, patients with an extended need for neurological rehabilitation per se precluded a return to the workforce were excluded. Even though the overall number of these patients were small, they could potentially have influenced our results with a lower likelihood of returning to the workforce. Thus, the current results represent a conservative estimate of a surgical population.

The study was also limited to patients between 18–63 years. We acknowledge, though, how several patients above the age of 63 were working and how they could have been included in the study. However, the age limit was set to ensure that no patients received state pension during the follow-up period.

Finally, the follow-up period of six months might be too short a period of follow-up, and with one-fifth of the patients still being on paid sick leave at the time of follow-up, future studies are encouraged to incorporate a more extended period of follow-up.

## Conclusion

In this cohort study, one-fifth of patients in the working-age were still on sick leave six months after open heart valve surgery. Age had little influence on the pattern of sick leave, except for the time of surgery. In regression analyses, prolonged sick leave prior to surgery and EuroScore ≥2.3 were associated with a lower likelihood of returning to the workforce. This knowledge might help identify patients at higher risk of detachment from the workforce and might be targeted during rehabilitation. In addition, the study adds to knowledge about what to expect following open heart valve surgery.

## Supporting information

**S1 Fig. Probability distribution of employment status six months before and after surgery among the total population of patients between 18–63 years.**
(PDF)

**S2 Fig.**
(PDF)

**S1 Table. Baseline characteristics of the total population (including patients on early retirement).**
(PDF)

**S2 Table. Patterns of sick leave and return to the workforce among the population being part of the workforce and divided by age groups.**
(PDF)

**S3 Table. Patterns of employment status before and after open heart valve surgery.**
(PDF)

# Author Contributions

**Conceptualization:** Britt Borregaard, Lars P. S. Riber.

**Data curation:** Britt Borregaard, Jordi S. Dahl, Jacob E. Møller.

**Formal analysis:** Britt Borregaard, Sasja M. Pedersen, Thomas P. H. Rothberg.

**Funding acquisition:** Britt Borregaard.

**Investigation:** Britt Borregaard, Jacob E. Møller.

**Methodology:** Britt Borregaard, Emil Fosbøl, Jacob E. Møller.

**Project administration:** Britt Borregaard.

**Resources:** Britt Borregaard.

**Supervision:** Jordi S. Dahl, Emil Fosbøl, Jacob E. Møller.

**Validation:** Britt Borregaard.

**Visualization:** Britt Borregaard.

**Writing – original draft:** Britt Borregaard.

**Writing – review & editing:** Britt Borregaard, Jordi S. Dahl, Ola Ekholm, Emil Fosbøl, Lars P. S. Riber, Kirstine L. Sibilitz, Sasja M. Pedersen, Thomas P. H. Rothberg, Maiken H. Nielsen, Selina K. Berg, Jacob E. Møller.

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
