## [Decision Letter · Decision Letter 0]

5 Aug 2020

PONE-D-20-20822

Employment status before and after open heart valve surgery: A cohort study

PLOS ONE

Dear Dr. Borregaard,

Thank you for submitting your manuscript to PLOS ONE. After careful consideration, we feel that it has merit but does not fully meet PLOS ONE’s publication criteria as it currently stands. Therefore, we invite you to submit a revised version of the manuscript that addresses the points raised during the review process.

We look forward to receiving your revised manuscript.

Kind regards,

Corstiaan den Uil

Academic Editor

PLOS ONE

Journal Requirements:

2. In ethics statement in the manuscript and in the online submission form, please provide additional information about the patient records used in your retrospective study. Specifically, please ensure that you have discussed whether all data were fully anonymized before you accessed them and/or whether the IRB or ethics committee waived the requirement for informed consent. If patients provided informed written consent to have data from their medical records used in research, please include this information.

4. Your ethics statement must appear in the Methods section of your manuscript. If your ethics statement is written in any section besides the Methods, please move it to the Methods section and delete it from any other section. Please also ensure that your ethics statement is included in your manuscript, as the ethics section of your online submission will not be published alongside your manuscript.

Reviewers' comments:

Reviewer's Responses to Questions

**Comments to the Author**

1. Is the manuscript technically sound, and do the data support the conclusions?

Reviewer #1: Partly

2. Has the statistical analysis been performed appropriately and rigorously? 

Reviewer #1: No

3. Have the authors made all data underlying the findings in their manuscript fully available?

Reviewer #1: Yes

4. Is the manuscript presented in an intelligible fashion and written in standard English?

Reviewer #1: Yes

5. Review Comments to the Author

Reviewer #1: Thank you for the opportunity to review this interesting manuscript. The research described in this paper seems to be a retrospective cohort study looking at the clinical predictors associated with returning to work after heart valve surgery in patients aged 18 to 63 years. The data used for the research come from hospital records and registry data (secondary data sources). The manuscript is written in a clear English that is easy to understand.

Overall, the study finds that poorer cardiovascular health (based on EuroScore-II score) and a longer hospital stay after the heart valve operation lowers the chance of returning to work. The authors claim that this is the first study to focus on clinical characteristics and return to work. I can believe this, because it is hard to see how this information, which is rather apparent and often a secondary finding of papers looking at rehabilitation interventions, merits research. Nevertheless, I have some comments and questions regarding the manuscript itself and the methods chosen that I hope will be helpful.

Abstract

Line 40: Maybe you could add a very brief description mention EuroScore as clinical characteristic in parentheses. I think a brief description of EuroScore would be helpful as well.

Line 47-48: I think it would be easier to read if the OR for of longer length of hospital stay was written in descending order.

Background

Lines 61-62: Other factors that might impact the ability to return to work after heart surgery are mentioned, without listing them. This is fine, but one factor that seems to be neglected in this paper is the type of work a patient will be returning to. Someone with a physical job may find it harder to return to work than someone with a desk job after heart surgery. This could be mentioned here and must be considered later in the statistical analysis.

Line 79: The second objective of the study is rather vague. The factors being investigated should be named. This seems to be an explorative study, as the objectives are formulated in a way that does not sound as if a concrete hypothesis is being tested. The fact that this is an explorative study should be stated in the paper.

Lines 91-97: Reading the list of exclusions, I got the impression these exclusions were decided while reviewing the medical records and not based on an a priori list of exclusions. Is this the case? Was an a priori study protocol prepared for this research (even one that was not published)?

Line 103: EuroScore is mentioned here very briefly. Later the tables state EuroScore-II. Please describe the EuroScore used, how it is calculated and the meaning of the index values here. Was the log of EuroScore-II really included in the regression models or the binary categories ≥2.3 vs <2.3?

As I understand it, the EuroScore is calculated prior to the operation. Are there any other clinical values, beside atrial fibrillation that describe the health of patients directly after the operation and can be considered?

Lines 116-121: This was the one part of the text I did not understand. I do not see how the three items listed in lines 116-118 line up with the descriptions provided in 119-120. Do items 1 and 2 correspond to the workforce described in lines 119? I think this needs to be rewritten to be clearer.

Who determined if someone was “still capable of working” for the DREAM registry? Is there a legal definition of “capable of working” that can be described here?

Lines 134-135: I do not see the use of doing testing for normality with the Shapiro-Wilks test. The regression model being used, does not require normality.

What is the time scale unit for return to work? Days? Weeks?

Logistic regression would not be my first or second choice for this type of data. Ideally this should be evaluated as time-to-event with either a Cox regression model. This allows for censoring of data, which means people dying or having a perioperative stroke after hospital discharge can be included into the analysis until the time of their death/stroke (at which time they are censored). If the time scale for return to work is not that exact, an alternative would be to calculated incidence rate ratios (IRR) using Poisson-regression and person-months as a time scale.

The longer one is away from work, the harder it may be to return. I think it would be interesting if the time on sick-leave (in days or weeks) just prior to the OP could be included in the analysis as a variable. While this might be a proxy for the severity of the underlying illness, prolonged time away from work before the OP may make it harder to imagine going back afterwards.

How was the model selection conducted?

Results

Line 152: Early retirement: I do not understand why the patients who were already retired before their operation are just now being excluded. Even though they are described as being excluded, they are included in table 2, which is confusing. I do not understand why their exclusion cannot be mentioned already in the methods and why they are not consistently excluded in the results. If you want to see if someone comes out of early retirement post-OP, this should be considered a new research question and considered separately in only the subgroup that was in early retirement pre-OP.

Table 1:

Was there any data on arterial blood pressure/ hypertension for the population that could be included in this table?

Did any of the patients take part in post-OP rehabilitation programs? If so, the prevalence of such rehabilitation should be described, and perhaps considered in the regression models as this might also impact return to work.

The type or field of work conducted pre-OP should be described. At least the proportion working in physically strenuous fields of work should be mentioned. Job strain would also be an interesting factor to consider. High job strain (e.g. demand control model) may also prevent return to work.

Were any of the patients self-employed? Could this partially explain the number of patients not on sick leave at the time of their operation? People who are self-employed tend to return to work sooner. This might also need to be considered in the regression model.

The youngest age range is wide: 18-45 years. I suspect the indications for the OP were different for the youngest patients, while coronary artery disease was probably more common in older patients. Can some information regarding the indications in the different age groups be provided?

Younger patients who were also students seem make it difficult to interpret the results for this youngest age group, as they would not be categorized as being on sick leave. Could students or the youngest patients (<30 years) be excluded from the analysis to check this? This could either replace the main analysis or be a sensitivity analysis.

Figures 2: This is an interesting depiction of the proportion on sick leave or working. Would it also be possible to depict the proportion of patients who go into early retirement post-OP?

Figure 3: The stratification of Figure 2 into the different age groups takes up a lot of space and adds little to the paper. I find it hard to compare the groups. think it might be nicer if the curves could be included in one single graphic. Maybe as Kaplan-Meier curves post-OP.

Figure 4: Although age is otherwise considered in age-groups, age seems to have been included as a continuous variable in the regression model. While this is legitimate, I think it would make it add to the interpretation of the descriptive analysis of return-to-work if the same age-categories were used in the regression model. I think suspect the chance of returning to work might also be significant for some age-groups, while the increased chance of returning to work is not significant for an increase in a single year of age. Also, increased chance of returning to work might not be linear for age. It might be lower for younger age groups, higher in the middle and lower again for the oldest age group. Using the age categories will show if this is the case.

Line 244 (and Figure 4): I do not think the term “co-morbidity” is the best descriptor to describe the EuroScore and post-OP atrial fibrillation. A better term for these indicators of cardiovascular health is needed. A co-morbidity would be additional diseases, such as diabetes or kidney disorders. Why were these not considered in the model?

6. PLOS authors have the option to publish the peer review history of their article (what does this mean?). If published, this will include your full peer review and any attached files.

Reviewer #1: No

---

## [Author Response · Author response to Decision Letter 0]

14 Sep 2020

Comment

Abstract

Line 40: Maybe you could add a very brief description mention EuroScore as clinical characteristic in parentheses. I think a brief description of EuroScore would be helpful as well.

Line 47-48: I think it would be easier to read if the OR for of longer length of hospital stay was written in descending order.

Answer

Thank you for the comments. We have changed the abstract according to the reviewer suggestion, but with the new results (see below)

Comment

Background

Lines 61-62: Other factors that might impact the ability to return to work after heart surgery are mentioned, without listing them. This is fine, but one factor that seems to be neglected in this paper is the type of work a patient will be returning to. Someone with a physical job may find it harder to return to work than someone with a desk job after heart surgery. This could be mentioned here and must be considered later in the statistical analysis.

Answer

Thank you to the expert reviewer for this comment. It is both an excellent and highly relevant suggestion. We have added a line about this in the background section. Unfortunately, the DREAM database used in the current study does not contain information on job type, why this data are not available.

Knowledge of specific factors associated with return to the workforce after open heart valve surgery is sparse: Unemployment one year before surgery has been associated with reduced likelihood of returning to the workforce [1], whereas participating in cardiac rehabilitation is not associated with returning to the workforce [12]. Some studies have investigated factors associated with the likelihood of returning to the workforce following coronary artery bypass grafting (CABG) and demonstrated how younger age, male sex, higher educational level, work status before surgery and higher income were associated with returning to the workforce [1, 13, 14]. Also, sick leave status before treatment and job type/strain are known to impact employment status following a heart disease [15]. Whether similar characteristics, including differences in age groups, are associated with returning to the workforce among patients undergoing open heart valve surgery, are currently unknown. This knowledge is important to identify patients at higher risk and thereby to prevent detachment from the workforce to the extent possible. 

Furthermore, we have elaborated in the limitations section. See below. 

Comment

Line 79: The second objective of the study is rather vague. The factors being investigated should be named. This seems to be an explorative study, as the objectives are formulated in a way that does not sound as if a concrete hypothesis is being tested. The fact that this is an explorative study should be stated in the paper.

Answer

Thank you for pointing this out – a very good perspective. Given the retrospective design based on national registries, no causalities can be demonstrated, and as the expert reviewer points out, the study should be interpreted as an explorative and instead hypothesis-generating. We have added this in the methods section

Comment

Lines 91-97: Reading the list of exclusions, I got the impression these exclusions were decided while reviewing the medical records and not based on an a priori list of exclusions. Is this the case? Was an a priori study protocol prepared for this research (even one that was not published)?

Answer 

The population was pre-defined and based on the following study (ClinicalTrials.gov NCT03053778). Thus, the exclusion criteria were given.

The current study was designed after the original study, but using the same cohort of patients and linked to register-based data. 

Comment

Line 103: EuroScore is mentioned here very briefly. Later the tables state EuroScore-II. Please describe the EuroScore used, how it is calculated and the meaning of the index values here. Was the log of EuroScore-II really included in the regression models or the binary categories ≥2.3 vs <2.3?

As I understand it, the EuroScore is calculated prior to the operation. Are there any other clinical values, beside atrial fibrillation that describe the health of patients directly after the operation and can be considered?

Answer

Thank you to the expert reviewer for this comment. We have elaborated on the section about the EuroScore II in the methods section. 

The EuroScore II is a logistic surgical risk evaluation calculated before surgery. We used the binary categories of this score in the regression models.

The text has been changed accordingly;

Demographic and clinical data

Demographic and clinical data were obtained from electronic medical records and the Western Denmark Heart Registry (WDHR) (16). Living status, smoking status, alcohol consumption, body mass index (BMI) and length of stay were obtained from the electronic medical records, whereas the type of surgery, co-morbidity and EuroScore II (surgical risk evaluation) were obtained from WDHR. The EuroScore II is a logistic surgical risk evaluation calculated before surgery including age, sex, renal impairment, extracardiac arteriopathy, poor mobility, chronic lung disease, active endocarditis, critical preoperative state, angina status, recent myocardial infarction, pulmonary hypertension, urgency and weight of the procedure (17).

As the EuroScore captures several other clinical variables / risk factors, we believed that this covers many of the suspected variables to be assumed related to our outcome. Also, due to lack of statistical power, we omitted further co-variates in the regression model. Thus, we restricted the number of variables in the multivariable models to avoid overfitting of models. This has been added to the statistics section.

Comment

Lines 116-121: This was the one part of the text I did not understand. I do not see how the three items listed in lines 116-118 line up with the descriptions provided in 119-120. Do items 1 and 2 correspond to the workforce described in lines 119? I think this needs to be rewritten to be clearer.

Who determined if someone was “still capable of working” for the DREAM registry? Is there a legal definition of “capable of working” that can be described here?

Answer 

Thank you to the expert reviewer for pointing this out. We have re-written the section, so hopefully, it is now more transparent and easier to follow:

Variables in the DREAM registry were grouped based on recommendations and similar study designs related to the workforce attachment into: 

1. On sick leave

2. Working/part of the workforce (employed, unemployed (but still capable of working based on the coding) or received paid leave of absence and educational grants)

3. Out of the workforce (patients were considered to be out of the workforce if they were on early retirement of any kind, n=65)

Comment

Lines 134-135: I do not see the use of doing testing for normality with the Shapiro-Wilks test. The regression model being used, does not require normality.

Answer 

The Shapiro-Wilks test was used to test for normality on continuous variables reported in the baseline table, and not, as the expert reviewer states, for the regression model.

Comment

What is the time scale unit for return to work? Days? Weeks?

Answer 

The time scale used for return to work was weeks, as the DREAM database includes changes in employment status weekly. This has not been highlighted.

Comment

Logistic regression would not be my first or second choice for this type of data. Ideally this should be evaluated as time-to-event with either a Cox regression model. This allows for censoring of data, which means people dying or having a perioperative stroke after hospital discharge can be included into the analysis until the time of their death/stroke (at which time they are censored). If the time scale for return to work is not that exact, an alternative would be to calculated incidence rate ratios (IRR) using Poisson-regression and person-months as a time scale.

Answer 

Thank you to the expert reviewer for letting us explain our choice of regression model.

As no patients died during the six-month follow-up (among patients being alive at discharge) and only a few patients changed their status from the workforce to out of the workforce, we did not find the Cox proportional the appropriate choice. This was discussed with the statistician in the group, and it was investigated whether there was a censoring problem (not a problem, a Cox did not change any results). Thus, the logistic regression model was chosen as the most appropriate model. 

Comment

The longer one is away from work, the harder it may be to return. I think it would be interesting if the time on sick-leave (in days or weeks) just prior to the OP could be included in the analysis as a variable. While this might be a proxy for the severity of the underlying illness, prolonged time away from work before the OP may make it harder to imagine going back afterwards

Answer 

Thank you for pointing this out. This is a great suggestion. We have incorporated patients with prolonged sick leave prior to surgery in the overall regression model. This has been added to the statistics section and results. 

Comment

Results

Line 152: Early retirement: I do not understand why the patients who were already retired before their operation are just now being excluded. Even though they are described as being excluded, they are included in table 2, which is confusing. I do not understand why their exclusion cannot be mentioned already in the methods and why they are not consistently excluded in the results. If you want to see if someone comes out of early retirement post-OP, this should be considered a new research question and considered separately in only the subgroup that was in early retirement pre-OP

Answer 

This is a very interesting comment, and we acknowledge how this can be confusing.

To accommodate the reviewer comment, we have moved table 2 to supplementary materials S3 Table. 

Also, we have re-structured the results sections to avoid further confusion.

Comment

Table 1:

Was there any data on arterial blood pressure/ hypertension for the population that could be included in this Table?

Did any of the patients take part in post-OP rehabilitation programs? If so, the prevalence of such rehabilitation should be described, and perhaps considered in the regression models as this might also impact return to work.

The type or field of work conducted pre-OP should be described. At least the proportion working in physically strenuous fields of work should be mentioned. Job strain would also be an interesting factor to consider. High job strain (e.g. demand control model) may also prevent return to work.

Answer 

We have added information about hypertension (a binary variable from our registry) to the Table. 

Unfortunately, we do not have information regarding participation in rehabilitation post OP, as this is only a requirement to report among patients with ischemic heart disease.

Also, an excellent comment about the type of work / job strain among the patients as seen above. This is, unfortunately, not information we have in our registry of employment status (DREAM). Instead, we have added something about this in the limitations section:

The use of the DREAM register is both the main strength and a limitation of the study; A strength as it covers all entries on sick leave compensations. As the entries are dependent on employers claiming compensation refund, the incentive for registration is high – ensuring high accuracy of data. The use of the register though, also includes limitations as only sick leave spells of >2 weeks are included. Thus, the register does not include short-term absence, and similarly, the register does not distinguish between full- and part-time sick leave [19, 20]. Similarly, as the register only include persons who receive a paid benefit, it can be difficult to conclude on the status of persons who are not registered. Type of job, job strain (e.g. physical strenuous fields), self-employment and similar knowledge would potentially also be relevant data of the population, but not available through the DREAM registry. This is also a limitation by the use of the DREAM register. The number of available jobs during the study period could potentially have had an impact on returning to work, why the main focus of the study was returning to the workforce. Classification of the groups; sick leave, workforce and out of the workforce could have caused a risk of classification bias but were performed as a method to classify DREAM data as previously described [13, 21, 22].

Comment

Were any of the patients self-employed? Could this partially explain the number of patients not on sick leave at the time of their operation? People who are self-employed tend to return to work sooner. This might also need to be considered in the regression model.

Answer 

See the answer above. This is not available information. Instead, we have added something about that in the limitations section. 

Comment

The youngest age range is wide: 18-45 years. I suspect the indications for the OP were different for the youngest patients, while coronary artery disease was probably more common in older patients. Can some information regarding the indications in the different age groups be provided?

Younger patients who were also students seem make it difficult to interpret the results for this youngest age group, as they would not be categorized as being on sick leave. Could students or the youngest patients (<30 years) be excluded from the analysis to check this? This could either replace the main analysis or be a sensitivity analysis

Answer 

Thank you for letting us explain this more clearly.

We acknowledge the perspective of the reviewer regarding surgical indications among different age groups – and yes, CAD were more common among the elderly patients compared to the younger groups. This perspective is partly captured in the EuroScore (weight of the surgery and recent MI).

The age groups were chosen to capture differences in the groups, and yes, although the range 18-45 is wide, the group size is similar to the next two. 

We have performed a sensitivity analysis, as suggested by the reviewer, where we have investigated the results of the regression analysis without patients on educational grants.

This has been added to both the method section, the results section and a supplementary figure 2. 

Comment

Figures 2: This is an interesting depiction of the proportion on sick leave or working. Would it also be possible to depict the proportion of patients who go into early retirement post-OP?

Answer 

We agree with the expert reviewer. This would be valuable information. But, as only a few patients go “out of workforce” during the follow-up period (see answer regarding the choice of regression model) it would be problematic to include those patients in the figure due to the Danish Law of sensitive personal data. 

Instead, we have added an overall figure of the total population, including follow-up as supplementary material (S1 Fig)

Comment

Figure 3: The stratification of Figure 2 into the different age groups takes up a lot of space and adds little to the paper. I find it hard to compare the groups. think it might be nicer if the curves could be included in one single graphic. Maybe as Kaplan-Meier curves post-OP

Answer 

Thank you to the expert reviewer for the great suggestion of a combined figure. As we want to capture the difference among age groups before surgery as well, we have combined the groups in a figure illustrating proportions of patients being on sick leave at different time points divided in age groups (new fig 3)

Comment

Figure 4: Although age is otherwise considered in age-groups, age seems to have been included as a continuous variable in the regression model. While this is legitimate, I think it would make it add to the interpretation of the descriptive analysis of return-to-work if the same age-categories were used in the regression model. I think suspect the chance of returning to work might also be significant for some age-groups, while the increased chance of returning to work is not significant for an increase in a single year of age. Also, increased chance of returning to work might not be linear for age. It might be lower for younger age groups, higher in the middle and lower again for the oldest age group. Using the age categories will show if this is the case.

Answer 

Thank you for letting us explain this. We acknowledge how age divided into age group would give us a more accurate picture of the changes in odds among patients at different age groups. 

Also, the reviewer is right; the change of returning to work is not linear with age. 

We have discussed the issue with the statistician of the group and changed the variables in the model: age is included as groups, as suggested, and instead, length of stay is included as a continuous variable (where the increased risk is linear). The text has been changed throughout the manuscript accordingly.

Comment

Line 244 (and Figure 4): I do not think the term “co-morbidity” is the best descriptor to describe the EuroScore and post-OP atrial fibrillation. A better term for these indicators of cardiovascular health is needed. A co-morbidity would be additional diseases, such as diabetes or kidney disorders. Why were these not considered in the model?

Answer 

Thank you to the expert reviewer for letting us change the wording of co-morbidity and the two variables. This is now changed in both the text and the figure.

As stated earlier in the review, we wanted to avoid over-fitting the model, why we included EuroScore as a combined marker of the surgical risk, instead of including the specific variables separately.

---

## [Editor Report · Decision Letter 1]

23 Sep 2020

Employment status before and after open heart valve surgery: A cohort study

PONE-D-20-20822R1

Dear Dr. Borregaard,

We’re pleased to inform you that your manuscript has been judged scientifically suitable for publication and will be formally accepted for publication once it meets all outstanding technical requirements.

Kind regards,

Corstiaan den Uil

Academic Editor

PLOS ONE
---

## [Editor Report · Acceptance letter]

28 Sep 2020

PONE-D-20-20822R1 

Employment status before and after open heart valve surgery: A cohort study 

Dear Dr. Borregaard:

I'm pleased to inform you that your manuscript has been deemed suitable for publication in PLOS ONE. Congratulations! Your manuscript is now with our production department. 

Kind regards, 

on behalf of

Dr. Corstiaan den Uil 

Academic Editor

PLOS ONE